# Calibration and Uncertainty Quantification for Single-Ended Raman-Based Distributed Temperature Sensing: Case Study in a 800 m Deep Coaxial Borehole Heat Exchanger

**DOI:** 10.3390/s23125498

**Published:** 2023-06-11

**Authors:** Willem Mazzotti Pallard, Alberto Lazzarotto, José Acuña, Björn Palm

**Affiliations:** Department of Energy Technology, Royal Institute of Technology (KTH), 10044 Stockholm, Sweden; alberto.lazzarotto@energy.kth.se (A.L.); jose.acuna@energy.kth.se (J.A.); bjorn.palm@energy.kth.se (B.P.)

**Keywords:** distributed temperature sensing, DTS, uncertainty, fiber optic, Raman, borehole, temperature, deep coaxial BHE, DTRT, confidence intervals

## Abstract

Raman-based distributed temperature sensing (DTS) is a valuable tool for field testing and validating heat transfer models in borehole heat exchanger (BHE) and ground source heat pump (GSHP) applications. However, temperature uncertainty is rarely reported in the literature. In this paper, a new calibration method was proposed for single-ended DTS configurations, along with a method to remove fictitious temperature drifts due to ambient air variations. The methods were implemented for a distributed thermal response test (DTRT) case study in an 800 m deep coaxial BHE. The results show that the calibration method and temperature drift correction are robust and give adequate results, with a temperature uncertainty increasing non-linearly from about 0.4 K near the surface to about 1.7 K at 800 m. The temperature uncertainty is dominated by the uncertainty in the calibrated parameters for depths larger than 200 m. The paper also offers insights into thermal features observed during the DTRT, including a heat flux inversion along the borehole depth and the slow temperature homogenization under circulation.

## 1. Introduction

Data-driven modeling and learning have become increasingly important in science and engineering. Along with them, the need for reliable measurements with well-characterized uncertainties has grown. The field of geothermal energy is no exception. On the contrary, underground inaccessibility and relatively scarce data make accurate measurements particularly vital to performing correct inference on existing or future systems.

For ground source heat pump (GSHP) systems in particular, knowledge of the underground temperature field is valuable for understanding the underlying heat transfer mechanisms in the ground. In turn, this combined knowledge can be used to improve the performance of the system in several ways, such as enhancing current performance, updating predictions about future operating conditions, and providing more information about underground thermo-hydraulic conditions.

One way to obtain information about the underground temperature field is to use distributed temperature sensing (DTS) inside boreholes. DTS enables temperature measurement with some spatial resolution along the sensor itself, a fiber optic cable, rather than in a specific location. Furthermore, these measurements can be performed quasi-simultaneously over the length of the fiber [1,2]. Selker et al. [3] discussed the pros and cons of several DTS techniques: fiber Bragg grating, Brillouin and Raman scattering. Most short-distance sensing (<15 km) is based on Raman optical time domain reflectometry (OTDR) [1].

Raman DTS systems have been used in various fields, including monitoring power transmission lines [4], fire detection and safety [5,6], volcanology [7,8], oil well monitoring [9], inferring paleoclimate [10,11,12], leakage detection in dams and dikes [13], hydrogeology [14,15], atmospheric science [16], and extraterrestrial geophysics [17].

In shallow boreholes the Raman DTS technique—simply referred to as DTS below—has been used for about three decades with first applications in hydrogeology [2,18,19]. In the reviewed literature, application of the DTS technique to GSHP research first appears in the context of distributed thermal response testing (DTRT) [20,21,22,23]. DTRTs based on DTS is still a current topic of research and even seems to have regained interest in recent years [24,25,26,27,28,29,30,31,32].

Besides their application to DTRTs, fiber optic cables have also been used to monitor the operation of borehole and aquifer thermal energy storages (BTES and ATES) [33,34,35,36,37,38,39,40]. Additionally, DTS data have indirectly been capitalized on to create thermal conductivity maps [41] or to measure the undisturbed temperature profile and evaluate its effect on design procedures [21,42,43,44].

In-situ DTS measurements are coupled to different challenges, three of which are presented hereafter. The first one, calibration, is described by Selker et al. [3]: “*an important aspect to keep in mind is that this is a technology that provides temperature data with minimal setup or interpretation required. It is necessary to calibrate the instrument to the cable by attaching the fiber-optic cable to the instrument with the cable in an environment of well-known temperature*”. When performing temperature measurements with a DTS device, in-situ calibration is indeed a requirement [1,45,46,47,48]. While commercial DTS units typically have an in-built calibration procedure, it may not be precise and accurate enough for hydrogeological applications [46,47,49].

For calibration and data acquisition with DTS, it is important to distinguish between single and double-end configurations. Single-ended configurations, in opposition to double-ended configurations, are setups in which the fiber optic cables are connected to the DTS instrument only through one end. For discussions on single versus double-ended configurations, the reader is referred to Tyler et al. [47], Hausner et al. [46], and Hausner and Kobs [50]. The present paper focuses on the calibration of single-ended configurations but calibration procedures for double-ended configurations may be found in [49,51,52]. Hausner et al. [46] discuss four different calibration methods for single-ended configurations and their application to two different case studies. The authors conclude that explicit calculation of calibration parameters leads to the best precision and accuracy. They also admit that this is challenging to achieve for field installations since at least two baths must be maintained at different temperature levels, which is especially demanding for long-term installations [53]. For the best calibration method, Hausner et al. report mean RMSEs of 0.131 K and 0.108 K for a laboratory and a field case study, respectively. The corresponding mean biases are of 0.016 K and 0.043 K, respectively. These values are calculated over sections of fiber in a validation bath. For the same fiber configuration, the manufacturer’s calibration leads to RMSEs statistics of 0.793 K and 0.583 K for the laboratory and field case studies, respectively, while bias statistics are of 0.792 K and −0.580 K. Notably, the RMSEs are dominated by the bias component for the manufacturer calibration. This suggests that RMSEs might be substantially reduced by removing biases through calibration. Lillo et al. [54] report calibration metrics in a similar way for five different calibration algorithms—including the one from [46]—and five different sites (of which one is a laboratory test) for single-ended duplexed configurations. They found an algorithm that further improves the calibration methods proposed in [46], although “*the calibration process does not necessarily fulfill physical considerations*”. Another important aspect that calibration may help tackle is the step losses caused by fusion splices, local strains, fiber damages, or tight bends [46,47,50].

The second challenge that may arise with in-situ DTS measurements is the temperature drifts caused by changing ambient conditions [3,46,47,48,50,55]. In particular, changing ambient temperature may lead to some disturbances in the internal oven of the DTS instrument used for the manufacturer’s calibration. Hence, the need for dynamic calibration is highlighted in the literature [53].

The third challenge with DTS measurements is the quantification of uncertainty, which is more complex in field setups and often still missing [44,55]. As an example, the word “uncertainty” does not appear once in one of the most recent and comprehensive reviews about Raman-based DTS with more than 500 reviewed references [56]. Tyler et al. [47] studied spatial and temporal repeatability, as well as spatial resolution for three different instruments (for a complete list of definitions of metrological terms in the field of fiber optic sensors, the reader is referred to Habel et al. [57] and to JCGM [58] for more general metrological definitions). For the temporal repeatability, the authors found standard deviations of 0.08, 0.13, and 0.31 K for biases of 0.33, 0.18, and 0.14 K for each instrument, respectively. For the spatial repeatability, standard deviations are of 0.02, 0.04 and 0.08 K and the biases are not given although it appears graphically that accuracy is better for the instruments with larger standard deviations. The authors report the use of a single calibration bath that was short (≈5 m) due to logistical challenges. They also characterized their temporal repeatability test as “*fairly short*” (2 h). For the spatial resolution, average values between 1.91 to 3.45 m were reported.

Simon et al. [59] provide an extensive investigation of spatial resolution with two different instruments in a laboratory setting. Three different methods for the estimation of the spatial resolution were compared. The correlation length (semivariogram) provided spatial resolution closest to the manufacturer’s specification while the 90% step change method and temperature spatial derivative method led to higher spatial resolutions. Notably, the 90% step method appears to be sensitive to local temperature gradients due to heat conduction along the fiber cable. One should therefore be mindful of extrapolating spatial resolution found with this method to the whole cable since temperature gradients may vary along its length.

Des Tombe et al. [52] propose a new calibration approach allowing for the quantification of uncertainty for the temperature. The temperature uncertainty solely accounts for the noise from the detectors and the uncertainty in the estimated parameters. They found a nearly constant 95% confidence bound of about ±0.3 K for an experiment with a 100 m fiber with a double-ended setup and three baths. Among other things, they observed that the contribution from the parameter estimation is small compared to the noise of the Stokes and anti-Stokes.

When DTS is applied for GSHP applications, it is in most cases unclear if calibration baths are used and how this affects the measurement uncertainty. In some cases, precision (a standard deviation value) is reported [20,27,37,45,60]. The reported precisions are in general in the order of 0.01–0.1 K, which is consistent with what is otherwise presented [3,47]. Accuracy is more seldom discussed. Monzó [61] found mean bias as high as 1.86 K for a 2.4 km long duplexed installation with six splices along the fiber length (not including the far-end splice). For the forward part of the duplexed configuration, the author found a maximum systematic error of −0.51 K. Abuasbeh and Acuña [33] report a maximum and mean systematic error of −0.19 K and 0.09 K, respectively, in a validation bath. Acuña et al. [20] theoretically evaluated the systematic error that could arise from the undetermined position of the fiber inside the borehole heat exchanger (BHE) under laminar flow conditions. They found a maximum possible bias of ±0.4 K for this effect. Fujii et al. [23] used the ±1.0 K accuracy and 0.1 K resolution specified by the manufacturer.

For the GSHP field, there is a lack of consistent metrics for characterization of uncertainty. As Wilke et al. [55] rightfully observe: “detailed information about the sensitivities, uncertainties and potential sources of errors of these instruments under test conditions and their impact on the results of advanced TRT are still missing”. Moreover, there are some specific issues with DTS measurements. For instance, one should keep in mind that the obtained temperatures are averages in time and space. As such, it seems needed to specify which sampling interval and temporal averaging are used when providing metrological metrics. Given a long enough integration time under repeatability conditions [58], one can obtain any arbitrarily low random error for the measured average temperatures. For a real case, however, this metric might not be the most relevant, especially if the dynamics are important. In such a case, the uncertainty of the underlying temperature—rather than its average—might be more relevant.

In the overall reviewed literature, metrological metrics are often presented for calibration baths (it should be noted that calibration baths are not the best metrics for accuracy since they are performed precisely to maximize accuracy within those baths), validation baths, or specific sections of fiber but it is not always clear how these metrics should be propagated to the rest of the fiber optic cable. Additionally, it is not always possible to perform an optimal calibration—i.e., a dynamic calibration with at least three different calibrated sections as suggested in [46]—e.g., for logistics constraints [44,47,53].

In this paper, a new calibration method for single-ended DTS configurations in a BHE with a single bath is proposed. In addition, a method for removing fictitious temperature drifts due to variations in ambient temperature is presented. Finally, and more importantly, the DTS measurement uncertainty was quantified for a DTRT in a 800 m deep borehole with coaxial BHE. Among other things, the uncertainty includes components from the Stokes/anti-Stokes signals, the calibration bath temperature and the parameter estimation used during calibration.

## 2. Materials and Methods

### 2.1. Measurement Principle

In most cases, when light is scattered by molecules, the scattering will be elastic so-called Rayleigh scattering. In other words, the scattered light will have the same frequency as the incident light. Sometimes, however, the scattered light will have a different frequency. When light is scattered by a molecule, the incident photon energy may turn into a phonon (a molecular quantum vibrational excitation). The scattered light would then have a lower frequency (lower energy) than that of the incoming light. This phenomenon is referred to as Stokes Raman scattering. On the other hand, the energy of a phonon may be captured by a scattered photon, meaning that scattered light would have a higher frequency. This is referred to as anti-Stokes. According to Bose–Einstein statistics, the ratio of Stokes/anti-Stokes intensities is proportional to [62,63,64,65]
(1)$$\frac{P_{S}}{P_{AS}}\propto e^{\frac{\Delta E}{k_{B}T}},\;$$
where *S* and AS refer to Stokes and anti-Stokes, ΔE=ℏΔω is the difference in energy related to the frequency shift between incident and scattered light, which also corresponds to the difference in the molecules vibrational energy levels (phonon); $k_{B}$ is the Boltzmann constant and *T* is the temperature. From Equation (Equation 1), it is clear that the Stokes/anti-Stokes signals may be used to infer local temperatures. Accounting for the fraction of the scattered light that reaches back to the instrument, the signal attenuation within the fiber (Beer’s law), the efficiency of the detector, and potential step losses, the temperature may be expressed as [46,65]
(2)$$T\left(z,t\right)=\frac{\gamma }{ln\left(\frac{P_{S}\left(z,t\right)}{P_{AS}\left(z,t\right)}\right)+C\left(t\right)-\Delta R\left(z\right)-\Delta \alpha z}\;,$$
where *z* is the distance from the instrument (inside the fiber), *t* is time, $\gamma =\frac{\Delta E}{k_{B}}$, *C* accounts for the fraction of light that is back-scattered to the instrument and the instrument detection efficiency, $\Delta R\left(z\right)=\sum_{n=1}^{N}\Delta R_{n}\theta \left(z-z_{n}\right)$ represents the potential (N) step losses, and Δα is the difference in the power attenuation coefficients for the Stokes and anti-Stokes signals. For more in-depth discussions about the Raman DTS measurement principle, the reader is referred to [62,63,65] and the supplementary material of [46].

### 2.2. In-Situ DTS Configuration

The experimental setup for this study is presented in Figure 1. It consisted of a Sensornet Halo-DTS${}^{TM}$ unit connected to two ca. 1050 m long multimode optical fibers (Solifos BRUsens temperature sensing cables) in a single-ended configuration. The fibers were inserted into a 800 m deep water-filled borehole equipped with a coaxial BHE (see [66] for more details on the BHE). The temperature measurements were performed during a heating DTRT. In this paper, distances inside the borehole may be referred to as borehole depth or length although there might be a difference between the two (if the borehole is unknowingly inclined for instance). The borehole length should not be confused with the actual distance along the fiber cable. The fiber cables were armored with stainless steel and placed in a nylon sheath to hold the water pressure. A fusion splice between connector patch cords and the fiber channels was placed at about 5 m fiber length from the DTS unit. The DTS unit had a 50 m long internal fiber coil used for dynamic adjustment of the parameter C(t) in Equation (Equation 2). This was carried out using a reference temperature sensor and a reference fiber section (−34.4 to −6.0 m) placed in an internal oven, i.e., in which air temperature was supposedly uniform. The internal fiber coil had a different differential loss term compared to the external fiber cables. This must be accounted for in Equation (Equation 2). Moreover, an assumed step loss was applied at the fiber–instrument connection (connector) and the step loss at the splice must be accounted for such that
(3)$$T\left(z,t\right)=\left\{\begin{array}{cc} \frac{\gamma }{ln\left(\frac{P_{S}\left(z,t\right)}{P_{AS}\left(z,t\right)}\right)+C\left(t\right)-\Delta \alpha_{int}\left(z+d_{int}\right)} & \mathrm{for}\;z<0.\\ \frac{\gamma }{ln\left(\frac{P_{S}\left(z,t\right)}{P_{AS}\left(z,t\right)}\right)+C\left(t\right)-\Delta R_{0}u\left(z\right)-\Delta R_{5}u\left(z-5\right)-\Delta \alpha z-\Delta \alpha_{int}d_{int}} & \mathrm{for}\;z\geq 0, \end{array}\right.\;,$$
where $\Delta \alpha_{int}$ and $d_{int}$ are the internal differential losses and coil length, respectively. Note that the connector is used as reference for the distance (z=0). The parameters C(t), $\Delta R_{0}$, and $\Delta \alpha_{int}$ can be found by reverse-engineering.

A calibration bath was placed right before the fiber cables went into the borehole. The bath was initially filled with water and ice and was well-mixed in order to have a uniform temperature of 0 °C in the bath. For both fiber cables, the length inside the calibration bath was about 46.7 m (between fiber distances of 194.8 m and 241.5 m for the first cable and between 207.0 and 253.7 m for the second cable). Figure 2 shows the Stokes and anti-Stokes lines for both fiber cables at about 3.3 h after the data acquisition start. Measurements were performed with an integration time of 5 min and a sampling interval of 2 m. The different sections (internal oven, patch cord, ambient air, borehole) and remarkable points (connector and fusion splice) are also shown in the figure.

Note that only one calibration bath was used. This is sub-optimal for calibration since there are usually three parameters to determine, so one would need at least three reference sections (three baths) [46]. In particular, having only one calibration bath may be challenging for the determination of the differential loss term Δα. For DTS application in boreholes, it is clearly not possible to place a calibration bath at the end of the fiber cable. Since there were four fiber channels in each fiber cable, we could have spliced the fiber onto itself, a so-called single-ended duplexed configuration. While this should be done whenever possible, it requires a special termination unit that can both hold the pressure and is small enough to fit in the BHE, which is not always practically or economically feasible. In addition, two extra splices would be introduced at the bottom which would add some complexity to the calibration procedure.

### 2.3. Correction of Temperature Drifts

As discussed in the introduction, changing ambient conditions may lead to fictitious temperature drifts in the DTS measurement [3,46,47,48,50]. Some units can manage larger variations of ambient conditions but this may still remain an issue. Dynamic calibration may be performed with the right configuration, although this is not always possible.

Here, the DTS unit was placed inside a container in which the conditions were not controlled. As a result, raw temperature data show a daily oscillation as seen in Figure 3. The figure shows the evolution of temperature at two different sections within the borehole (around 200 and 700 m) during the different phases of the DTRT: undisturbed ground, circulation, heat injection, and heat recovery. It is quite clear that there is an oscillating pattern throughout time. Now, the question is: is this oscillation real or is it fictitiously induced by the measuring device? During the heat injection phase of the DTRT—between about 20 and 230 h—it is unlikely that these fluctuations are real, but they cannot be completely excluded since the water being circulated inside the BHE could be influenced by above-ground ambient conditions (e.g., these oscillations were also noticed in the sections of the fiber located outside the borehole).

During the heat recovery phase, however—from about 230 h until the end of the test—it is highly unlikely that these fluctuations are real because the water circulation is stopped, so there is no reason to think that the ambient conditions above ground would affect the temperature 200 m, let alone 700 m, below ground. This can be therefore be used as a condition for removing these oscillations. Since γ is constant, and since the step losses and differential losses are considered time-independent, the potential correlation can be determined to a good approximation using the inverse of the temperature (see Appendix A). Hence, the correction may be expressed as $\beta \left(T_{ref}\left(t\right)-{\bar{T}}_{ref}\right)$, where β is the regression coefficient such that
(4)$$\beta \left(T_{ref}\left(t\right)-{\bar{T}}_{ref}\right)=-\frac{1}{N_{Z_{bh}}}\sum_{z\in Z_{bh}}ln\left(\frac{P_{S}\left(z,t\right)}{P_{AS}\left(z,t\right)}\right)-C\left(t\right),\;t\in T_{rec}.$$

This correction is added to the denominator of Equation (Equation 2). Here, $Z_{bh}$ is taken as the whole borehole starting from 15 m below the surface to make sure that any influences from the above-ground temperature are excluded. $N_{Z_{bh}}$ is the number of measurement sections within $Z_{bh}$ while the time span $T_{rec}$ is taken as the recovery period, between 261 and 355 h, since the water circulation is stopped during that period. Undisturbed conditions can also be used, given a long enough period of measurements (e.g., if the recovery period is missing).

### 2.4. Calibration

As noted above, there was only one calibration section for each fiber. This is a sub-optimal configuration for determining the unknown parameters in Equation (Equation 2), i.e., γ, C(t), ΔR(z) and Δα. The first term, γ, is mostly dependent on the DTS instrument. Thus, it is reasonable to use the default value, thereby having one less parameter to estimate. In addition, C(t) was already calibrated by the instrument and corrected according to the previous section. Hence, this parameter may also be removed from the parameter estimation list.

Hence, only ΔR(z) and Δα remained to be estimated. Here, the temperature between the connector and the fusion splice (0–5 m) is not of interest so it was sufficient to use the default step loss value at the connector and calibrate the step loss at 5 m. Another way to state this is that the step losses were aggregated into one value only with the consequences that temperatures between 0 and 5 m of fiber cable may not be inaccurate. Accordingly, the parameter estimation problem turns into a simple linear regression with the fiber length as a variable,
(5)$$\Delta \alpha \cdot z+\Delta R=\frac{1}{N_{T_{bath}}}\sum_{t\in T_{bath}}\left(ln\left(\frac{P_{S}\left(z,t\right)}{P_{AS}\left(z,t\right)}\right)+C\left(t\right)\right)-\frac{\gamma }{T_{bath}}-\Delta \alpha_{int}d_{int},\;z\in Z_{bath}.$$

The bath sections used for calibration, $Z_{bath}$, were taken as 198.9–237.5 m for the first fiber cable and 211.0–249.6 m for the second cable. That is, they were taken as slightly shorter than the total bath section to avoid any temperature bias due to spatial resolution where the fiber enters and exits the bath [57]. The temperature of the bath, $T_{bath}$, was maintained at 0 °C between 0.5 and 6.85 h after the start of the data acquisition ($T_{bath}$). Potential variations in this temperature are accounted for in the uncertainty analysis.

The calibration was performed for each fiber cable separately, although—as noted in previous sections—it was expected that this will give a poor estimation of Δα since the linear regression of Equation (Equation 5) is performed over a limited section of fiber. Another approach is to combine the calibration of the two fiber cables by stating that the bottom temperature should be equal for both the cables at all times. This leads to the following equation:(6)$$\left(\Delta \alpha_{1}-\Delta \alpha_{2}\right)z+\Delta R_{1}-\Delta R_{2}=\frac{1}{N_{T}}\sum_{t\in T}\left(ln\left(\frac{P_{S,1}\left(z,t\right)\cdot P_{AS,2}\left(z,t\right)}{P_{AS,1}\left(z,t\right)\cdot P_{S,2}\left(z,t\right)}\right)+C_{1}\left(t\right)-C_{2}\left(t\right)\right),\;z\in Z_{bottom},$$
where the subscripts 1 and 2 refer to the first and second fiber cables (or channels). The bottom section, $Z_{bottom}$, could just be taken as the bottom-most measurement section although that risks giving too much importance to measurement noise at this length. Instead, a slightly longer section may be considered if the temperatures in the two legs of the BHE are known to not vary significantly (as at the bottom of a deep borehole for instance). Note that T represents the whole test period. Potential deviations from temperature equality at the borehole bottom were investigated with the uncertainty analysis.

The previous expressions may be combined in a compact block matrix form:(7)$$\left[\begin{array}{cc} X_{1} & 0\\ 0 & X_{2}\\ X_{12_{1}} & -X_{12_{2}} \end{array}\right]\left[\begin{array}{c} \Delta R_{1}\\ \Delta \alpha_{1}\\ \Delta R_{2}\\ \Delta \alpha_{2} \end{array}\right]=\left[\begin{array}{c} Y_{1}\\ Y_{2}\\ Y_{12} \end{array}\right]\;,$$
where, for $p\in \left\{1,2\right\}$
(8)$$X_{p}=\left[\begin{array}{cc} 1 & z_{p,1}\\ \vdots & \vdots \\ 1 & z_{p,n} \end{array}\right]Y_{p}=\left[\begin{array}{c} y_{p,1}\\ \vdots \\ y_{p,n} \end{array}\right]\;X_{12_{p}}=\left[\begin{array}{cc} 1 & z_{p,m-q}\\ \vdots & \vdots \\ 1 & z_{p,m} \end{array}\right]Y_{12}=\left[\begin{array}{c} y_{12,m-q}\\ \vdots \\ y_{12,m}. \end{array}\right]$$

The integer *n* corresponds to the n-th section in the calibration bath, while *m* corresponds to the section at the borehole bottom and *q* indicates the number of extra sections taken into the equality condition expressed in Equation (Equation 6). For instance, if *q* equals 0, only the bottom measurement section is considered while if *q* equals 1, the equality condition is applied to both the last and penultimate sections.

For $p\in \left\{1,2\right\}$ and $l\in \left[1,n\right]$, the terms $y_{p,l,k}$ of the vector $Y_{p}$ may be expressed as the right-hand side of Equation (Equation 5),
(9)$$y_{p,l}=-\frac{\gamma }{T_{bath}}+\bar{ln\left(\frac{P_{S_{p,l}}}{P_{{AS}_{p,l}}}\right)}-\Delta \alpha_{int}d_{int}+\bar{C_{p}}.$$

Similarly, for $l\in \left[0,q\right]$, the terms $y_{12,m-l,k}$ of the vector $Y_{12}$ may be expressed as the right-hand side of Equation (Equation 6),
(10)$$y_{12,m-l}=y_{1,m-l-y_{2,m-l}}=\bar{ln\left(\frac{P_{S_{1,m-l}}}{P_{{AS}_{1,m-l}}}\frac{P_{{AS}_{2,m-l}}}{P_{S_{2,m-l}}}\right)}+\bar{C_{1}}-\bar{C_{2}},$$
where the averages in the two previous equations are with respect to time.

### 2.5. Uncertainty Evaluation

#### 2.5.1. Propagation of Error

Once the calibration was performed, a natural question that arose was that of the uncertainty. The temperature uncertainty was determined using the state-of-the-art definitions, methods, and procedures described in [58], in particular regarding the propagation of uncertainty (combined uncertainty). The temperature measurement can be modeled by Equation (Equation 3) and the linear regression for the calibration expressed in Equations (Equation 5)–(Equation 7). Thus, the relative temperature uncertainty may be expressed as
(11)$$\begin{array}{cc} \frac{u_{c}\left(T\right)}{T}=\frac{T}{\gamma }( & u^{2}\left(ln\left(\frac{P_{S}}{P_{AS}}\right)\right)+u^{2}\left(C\right)+\left(T_{ref}-{\bar{T}}_{ref}\right)u^{2}\left(\beta \right)+u^{2}\left(\Delta R_{lin.reg.}\right)\\ \; & +z^{2}\cdot u^{2}\left(\Delta \alpha_{lin.reg.}\right)+2z\cdot u\left(\Delta R_{lin.reg.},\Delta \alpha_{lin.reg.}\right){)}^{1/2}. \end{array}$$

Note that the combined uncertainty, $u_{c}\left(T\right)$, depends on both the distance and time—as does the temperature. The term u(X) stands for the standard uncertainty of quantity *X*.

The standard uncertainty related to the Stokes and anti-Stokes was determined using a condition when the underlying temperature is expected to be constant. Then, all the observed variations in temperatures were attributed to variations in the measured Stokes and anti-Stokes signals such that
(12)$$u\left(ln\left(\frac{P_{S}}{P_{AS}}\right)\right)=u\left(T_{DTS}\right)\frac{\gamma }{T_{DTS}^{2}}.$$

There may be other effects on the temperature reading during the chosen period (e.g., variation in the internal oven temperature and its influence on *C* or actual variations due to fluid movement inside the BHE) but these effects are considered small since the period is short. Moreover, even if such effects are present, the main consequence would be that the uncertainty attributed to the Stokes and anti-Stokes readings was slightly overestimated, hence leading to a somewhat conservative evaluation of uncertainty. It must be clear that the uncertainty of Equation (Equation 12) is the uncertainty of the Stokes and anti-Stokes averaged over the chosen integration period (here five minutes) and not the uncertainty of every single Stokes and anti-Stokes signal measured by the instrument within that period (before internal processing). Note that the temperature directly obtained from the instrument, $T_{DTS}$, was used. This was to avoid introducing any other type of error into the standard uncertainty.

Using Equation (Equation 3) (z<0), the internal oven and its reference temperature, one can compute different values for the parameter C(t) over the whole internal reference section. The standard uncertainty of the parameter *C* is then simply taken as the standard deviation of the different *C* samples within the internal reference section.

The standard uncertainty of the correction term may be computed from the standard deviation of the residuals, ${\hat{\sigma }}_{\varepsilon }$, as:(13)$$u\left(\beta \right)=\frac{{\hat{\sigma }}_{\varepsilon }}{\sqrt{\sum_{t\in T_{rec}}{\left(T_{ref}\left(t\right)-{\bar{T}}_{ref}\right)}^{2}}}=\sqrt{\frac{1}{\left(N_{T_{rec}}-1\right)}\frac{\sum_{t\in T_{rec}}{\hat{\varepsilon }}^{2}\left(t\right)}{\sum_{t\in T_{rec}}{\left(T_{ref}\left(t\right)-{\bar{T}}_{ref}\right)}^{2}}}.$$

This assumes that the residuals are i.i.d. The uncertainty components related to the linear regression coefficients may be estimated in a similar way, i.e., through the covariance matrix, Σ, of coefficient estimates.
(14)$$\Sigma ={\hat{\sigma }}_{\varepsilon }^{2}{\left(X^{T}X\right)}^{-1},$$
where the matrix X is the leftmost block matrix of Equation (Equation 7). The variance terms of the linear regression coefficients are on the diagonal while the rest of the matrix terms are covariances. Note that ${\hat{\sigma }}_{\varepsilon }$ is a generic term for the standard deviation estimate of the residuals but it is not the same in Equations (Equation 13) and (Equation 14).

#### 2.5.2. Deviations in Bath and Bottom Conditions

Up to this point, it has been assumed that the bath temperature was fixed at 0 °C and that there was a perfect temperature equality between the two fiber cables at the bottom of the borehole. While these are most reasonable assumptions to make, it is not inconceivable that small deviations in those conditions might exist. How would such small deviations impact the calibration and the temperature uncertainty? This question cannot be answered simply using the propagation of error method. Thus, a Monte Carlo study was performed in order to answer that question. Two random variables were used for the study such that
(15)$$\left\{\begin{array}{c} T_{bath}^{\prime }∼N\left(T_{bath},\sigma_{bath}^{2}\right)\\ T_{2}\left(z,t\right)=\psi T_{1}\left(z,t\right)\;\forall \left(z,t\right)\in \left(Z_{bottom},T\right)\;,\;with\;\psi ∼N\left(1,\sigma_{bottom}^{2}\right) \end{array}\right..$$

Note that the random variables are consistent in time, meaning that they behave as systematic errors and are not re-sampled at every time step. The temperature $T_{bath}^{\prime }$ is the the actual bath temperature, while $T_{bath}$ is the temperature that is most reasonable to assume in the absence of any other information. Here, the latter was taken as 0 °C (273.15 K) while the standard deviation $\sigma_{bath}$ was assumed to be 0.1 K. $T_{1}$ and $T_{2}$ refer to temperatures from the first and second channels, respectively. The standard deviation of the ratio was assumed to be 0.05%. As an example, a $T_{1}$ equal to 20 °C (293.15 K) leads to a standard deviation of the difference of about ±0.15 K. The potential deviation at the bottom is written as a ratio rather than as an additional correction so that the linear regression of Equation (Equation 7) may still be applied with a small modification:(16)$$\left[\begin{array}{cc} X_{1} & 0\\ 0 & X_{2}\\ X_{12_{1}} & -X_{12_{2}} \end{array}\right]\left[\begin{array}{c} \Delta R_{1}\\ \Delta \alpha_{1}\\ \psi \Delta R_{2}\\ \psi \Delta \alpha_{2} \end{array}\right]=\left[\begin{array}{c} Y_{1}\\ \psi Y_{2}\\ Y_{12}\left(\psi \right) \end{array}\right],$$
where the terms $y_{12,m-l}$ of vector $Y_{12}\left(\psi \right)$ are expressed as
(17)$$y_{12,m-l}=\bar{ln\left(\frac{P_{S1,m-l}}{P_{AS1,m-l}}\right)}+\bar{C_{1}}-\psi \left(\bar{ln\left(\frac{P_{S2,m-l}}{P_{AS2,m-l}}\right)}+\bar{C_{2}}\right).$$

The estimated parameters for the second fiber may then be corrected by 1/ψ in order to find the parameters of interest. The Monte Carlo study was performed for the bath condition only, for the bottom condition only, and for a combination of both. This was undertaken to identify the contribution of each condition. Thus, the estimated parameters may be decomposed into different terms, such that
(18)$$\left\{\begin{array}{c} \Delta R=\Delta R_{lin.reg.}+\delta R_{bottom}+\delta R_{bath}\\ \Delta \alpha =\Delta \alpha_{lin.reg.}+\delta \alpha_{bottom} \end{array}\right.,$$
where δ symbolized random components resulting from the Monte Carlo study. Note that the bath condition only affects the intercept terms in the linear regression of Equations (Equation 5) and (Equation 7) since it just acts as an offset for the y’s terms. In terms of the uncertainty, this leads to six extra variance and covariance terms,
(19)$$\begin{array}{cc} \frac{u_{c}\left(T\right)}{T}=\frac{T}{\gamma }( & u^{2}\left(ln\left(\frac{P_{S}}{P_{AS}}\right)\right)+u^{2}\left(C\right)+\left(T_{ref}-{\bar{T}}_{ref}\right)u^{2}\left(\beta \right)\\ \; & +u^{2}\left(\delta R_{bottom}\right)+u^{2}\left(\delta R_{bath}\right)+2u\left(\delta R_{bottom},\delta R_{bath}\right)\\ \; & +z^{2}u^{2}\left(\delta \alpha_{bottom}\right)+2z\cdot u\left(\delta R_{bottom},\delta \alpha_{bottom}\right)+2z\cdot u\left(\delta R_{bath},\delta \alpha_{bottom}\right)\\ \; & +z^{2}\cdot u^{2}\left(\Delta \alpha_{lin.reg.}\right)+u^{2}\left(\Delta R_{lin.reg.}\right)+2z\cdot u\left(\Delta R_{lin.reg.},\Delta \alpha_{lin.reg.}\right){)}^{1/2}. \end{array}$$

#### 2.5.3. Temperature Averaging

So far, the uncertainty has been expressed for the temperature that is averaged during a user-chosen integrating time. As discussed in the introduction, the uncertainty of the average temperature becomes arbitrarily low for a long enough integration time. The underlying true temperature will, however, most likely change during the integrating time. If the underlying temperature is the actual measurand of interest, the uncertainty related to the temperature dynamics under the integration period should therefore be taken into account. The temperature at any time t+Δt between two integrating times *t* and $t+\Delta t_{int}$ may be decomposed into
(20)$$T\left(z,t+\Delta t\right)={\bar{T}}_{\Delta t_{int}}\left(z\right)+\Delta T\left(z,t+\Delta t\right).$$

The temperature dynamics obviously depends on the application and conditions specific to each measurement. It is therefore hard to say something general about the error term ΔT(z,t+Δt). Nevertheless, assuming that the integrating time is short enough so that the temperature may be approximated as a piece-wise linear function in the time domain, the error term ΔT(z,t+Δt) may be expressed as
(21)$$\Delta T\left(z,t+\Delta t\right)=\frac{\partial T}{\partial t}\left(z,t\right)\left(\Delta t-\frac{\Delta t_{int}}{2}\right).$$

Thus, the error at a given time and distance under such conditions may be treated as a random variable such that
(22)$$\Delta T∼U\left(-\left|\frac{\partial T}{\partial t}\right|\frac{\Delta t_{int}}{2},\frac{\partial T}{\partial t}\frac{\Delta t_{int}}{2}\right).$$

In that case, the uncertainty related to averaging may be expressed as
(23)$$u\left(\Delta T\right)=\left|\frac{\partial T}{\partial t}\right|\frac{\Delta t_{int}}{\sqrt{12}}=\frac{\left|\Delta T_{max}\right|}{\sqrt{12}}.$$

Not surprisingly, the standard uncertainty related to averaging is directly proportional to the first-order derivative (or maximum temperature change). For TRTs, this means that the uncertainty will be largest when the heat injection starts or stops and will quickly decrease after. According to data in [67,68], the standard uncertainty is always below 0.1 K after the first hour of heat injection.

#### 2.5.4. Overall Uncertainty, Degrees of Freedom and Extended Uncertainty

The standard uncertainty related to averaging and the standard uncertainty of Equation (Equation 19) may be combined to compute the overall standard uncertainty such that
(24)$$u_{c}\left(T\right)=\sqrt{u_{c}^{2}\left(\bar{T}\right)+u^{2}\left(\Delta T\right)}.$$

This standard uncertainty may then be extended to any given confidence level as
(25)$$U\left(T\right)=t_{1-\frac{\zeta }{2},\nu_{eff}}u_{c}\left(T\right)\;,$$
where $t_{1-\frac{\zeta }{2},\nu_{eff}}$ is the $1-\frac{\zeta }{2}$ quantile of the Student t distribution with $\nu_{eff}$, the effective degrees of freedom. The effective degrees of freedom is calculated from the Welch–Satterthwaite formula as reported in [58].

## 3. Results

### 3.1. Correction of Temperature Drifts and Calibration

Temperatures resulting from the calibration—including the correction of temperature drifts—are shown in Figure 4. The figure shows temperature profiles in time at different depths and for the two fiber channels. Besides the raw data, the results from three calibration methods are shown: on-site calibration using the instrument software, segregated calibration (i.e., each channel is calibrated on its own without the bottom temperature equality condition of Equation (Equation 6)), and combined calibration. Segregated and combined calibration methods also include the correction of temperature drifts.

The raw data and on-site calibration show daily oscillations that are typical of temperature drifts in the internal oven. As mentioned earlier, it is unlikely that such daily variations are real in the borehole, especially during the recovery phase when heat and circulation are turned off.

Another issue is the inequality in temperature between the two channels during the recovery period (from about 230 h). This happens for the raw data, the on-site calibration, and the segregated calibration, for which the temperature inequality is particularly large. Near temperature equality is expected since both heat and circulation are turned off during heat recovery.

The combined calibration remedies both of these issues. It also leads to matching temperatures at the bottom of the borehole—by definition—which can be seen in Figure 5. It is also an encouraging indication that the undisturbed temperature is very similar for both channels and that the temperatures at the top of the borehole are nearly equal during circulation.

The parameters obtained through the different calibration procedures are given in Table 1, while Table 2 gives the fixed parameters during calibration, including the temperature drift correction β. The default differential loss and that found for the combined calibration are similar. However, not accounting for the step loss due to the fusion splice leads to a significant difference in temperatures as seen in Figure 5. It is noteworthy that the temperature profiles from the raw data match at the bottom of the borehole, without any prior constraints unlike in the combined calibration procedure. On the other hand, the segregated calibration does not lead to satisfactory results at the bottom of the borehole. This is mostly due to the large differences in differential loss terms, Δα, which themselves arise from the linear regressions over a small section (calibration bath) compared to the whole fiber length. The on-site calibration leads to a somewhat better temperature match at the bottom. In this case, it would have been better to solely calibrate the step loss, ΔR, instead of calibrating both the differential and step losses.

#### Verification of the Calibration

To completely validate the proposed calibration procedure, an independent measurement would be required. This was unfortunately not available for this site. Nevertheless, another DTS measurement in the same borehole was performed about one year prior to the measurements previously described. The same instrument and fiber cables were used, although the channels were switched. Applying the same calibration procedure, the temperature profiles shown in Figure 6 were obtained.

The RMSEs over the whole borehole length (top 15 m excluded to avoid influences from air temperatures) are about 0.18 and 0.16 °C for the first and second fibers, respectively (i.e., the first and second channels as presented in this paper). In addition, the values found for the differential loss term are very similar for the first and second fiber cables, which is expected since the two cables are the same product.

From both the visual comparison in Figure 6 and the numerical RMSE indicators, the calibration procedure appears to be consistent for the two different measurements. The raw data show, for instance, a much weaker consistency as can be seen in Figure 6.

Nevertheless, the estimated parameters appear to be different from one year to another as shown in Table 1. For ΔR, this is expected since disconnecting and re-connecting fiber cables to the DTS instrument will lead to different step losses. For the differential losses Δα, however, the difference is more surprising because this parameter is supposedly only dependent on the fiber cable itself. In the next section, the uncertainty bounds for Δα will help clarify this issue.

### 3.2. Uncertainty Evaluation

Using the methodology developed in Section 2.5, the temperature uncertainty may be computed for all times and distances. The uncertainty in time appears to be relatively constant as can be seen in Figure 7, which shows the calibrated temperature profiles vs. time at the same four depths as shown in Figure 4. The figure hints at an increasing temperature uncertainty with increasing distance. This is confirmed by Figure 8, where vertical temperature profiles in the borehole are shown together with their 95% confidence bounds at three different times. Here again, the chosen times are the same as in Figure 5.

On average, the confidence bounds reach a minimum at around 10 m depth, probably due to influence from the top boundary effects that are reflected in the Stokes/anti-Stokes uncertainty component at the top of the borehole. Except for the topmost 10 m, the 95% confidence bounds show an increasing, non-linear trend as the distance increases (or borehole length). For channel 1, the expanded uncertainty increases from around 0.35 K at 10 m to about 1.65 K at the bottom, while it is from around 0.4 K to 1.65 K for channel 2. At around 300 m, the expanded uncertainty is about 0.74 K for both channels. Values of the expanded uncertainty for different depths are shown in Table 3 (for a sampling interval of 2.029 m and a temporal averaging of 5 min. The effective degrees of freedom vary with depth but is, on average, between 40 and 180).

The fact that the two expanded uncertainties are very similar can be seen as a sign of robustness. The manufacturer’s temperature resolution [69] is also given in Table 3 although the lack of clarity around the term makes it hard to compare with the uncertainty figures. According to [52], the temperature resolution can be understood as a standard uncertainty, in which case the uncertainty bounds found in this study are about one degree of magnitude higher than what is reported in product data sheets.

The contribution from the different components of the combined uncertainty is shown in Figure 9 for both channels. Note that the figure shows the combined variance instead of the combined standard uncertainty. This is because the different uncertainty components add up in quadrature instead of being a simple addition. Thus, simply stapling the standard uncertainty of each component would not be indicative of their contribution to the combined uncertainty. The combined variance and its different components are shown for a given time (300 h) but they are close to constant in time so the figure is representative of other moments.

The linear regression component clearly dominates the combined variance for larger depths while it almost vanishes close to the borehole top. This is expected: close to the calibration bath, the calibration uncertainty will be lower and vice-versa. For the same reason, it makes sense that the variance related to potential systematic errors in the bath temperature also increases with the distance, although it is not noticeable in the figures. Similarly, for potential systematic errors in the bottom equality condition, the contribution is small but can be seen for a small section close to the borehole bottom in the zoomed areas at the top of the two sub-figures. The lower zoomed areas show that the contribution is vanishingly small at shallow depths. Somewhat unexpectedly, the contribution from the depth-constant C(t) shows a depth-dependency. The variance $u^{2}\left(C\right)$ is constant for a given time—as expected—but the latter is multiplied by $\frac{T^{4}}{\gamma^{2}}$ (see Equation (Equation 19)) to obtain the contribution to the temperature-combined variance, which creates the depth-dependency. The contribution from both the correction based on the internal temperature and the averaging is negligibly small. The former is not even visible on any of the zoomed areas of Figure 9. The latter is even negligible during periods in which temperature changes are higher—e.g., at the beginning of the heat injection.

With regard to the previous discrepancy found between the differential loss term during the two calibrations mentioned in the previous section: the found 95% confidence bounds for the differential loss of channel 1 are $9.4739\pm 1.172\times 10^{-5}$ m${}^{-1}\;\left(\nu =40\right)$ and $8.4853\pm 1.160\times 10^{-5}$ m${}^{-1}\;\left(\nu =25\right)$, while they are $9.4630\pm 1.190\times 10^{-5}$ m${}^{-1}\;\left(\nu =40\right)$ and $8.6884\pm 1.146\times 10^{-5}$ m${}^{-1}\;\left(\nu =25\right)$ for channel 2. Hence, the values found during the two different calibrations are compatible with one another. The difference may be due to a shorter calibration section in the independent calibration. Note that covariance with the step losses is not considered since the latter effectively changes from one calibration to the other—due to the disconnection and re-connection of the fiber cables.

## 4. Discussion

DTS offers interesting perspectives in BHE and GSHP applications and has gained interest in recent years, be it for monitoring, field tests (DTRTs), validation of heat transfer models, estimation of the geothermal gradient, or evaluation of undisturbed temperature or thermal conductivity maps [21,24,25,26,27,28,29,30,33,34,35,36,37,41,42,43,44,54].

In this study, a new calibration method and the quantification of uncertainty for single-ended Raman-based DTS were proposed, as well as a correction for fictitious temperature drifts caused indirectly by ambient temperature variations. The method was developed to address sub-optimal calibration setups since those are fairly common in BHE applications. For instance, it is not always possible to use the minimum recommended of two calibration baths [46] and field conditions may lead to systematic errors (e.g., due to ambient temperature variations). In addition, the far-end of the fiber cable is not usually accessible (since the fiber cable is inside the borehole and not spliced at the bottom). This makes the calibration of the differential loss term Δα more challenging. The proposed calibration method is based on one calibration bath only and uses temperature equality between the two fiber cables as an extra constraint. In addition, a correction for fictitious temperature oscillation induced by variation of ambient temperature was proposed. This correction is more specific to measurements in which the DTS instrument is subjected to significant temperature variations—typically outdoor field measurements with non field-customized DTS units. Finally, a thorough quantification of temperature uncertainty was proposed. The combined uncertainty notably includes uncertainty components from the calibration method (bottom equality condition, calibration bath temperature, linear regression), the correction for temperature oscillation, and the built-in temperature averaging.

The three elements listed above were tested for a field application in a 800 m deep borehole equipped with a coaxial BHE. One fiber cable was placed in the annulus of the heat exchanger while another cable was placed in the center pipe. A single calibration bath was used for the two fiber cables.

The proposed calibration method leads to more consistent results than raw data, segregated calibration for each cable, or on-site calibration using the instrument interface. The consistency is better between the two fiber cables—they show the same temperature under undisturbed conditions and heat recovery—but also in time—the calibration leads to similar temperature profiles when performed in a different setup with about one year in between. Nonetheless, the different values for the differential loss between the two calibrations at a one-year interval should be investigated further, even though they are not being strictly inconsistent according to the found uncertainty bounds.

The proposed correction of temperature drifts due to the influence of ambient air temperature variations successfully filters out oscillations from the raw data. The oscillations were filtered out for the whole test period even though the correction was only based on times at which there is, a priori, no reasonable correlation between the ambient air temperature and the temperature at the bottom of the borehole—typically, undisturbed conditions or heat recovery without circulation.

At a given depth, the temperature uncertainty is consistent in time with 95% confidence bounds of ±0.45, 0.58, 0.74, 0.90, 1.1, 1.3, 1.5, and 1.7 K at 100, 200, 300, 400, 500, 600, 700, and 800 m depths, respectively (for a sampling interval of 2.029 m and a temporal averaging of 5 min. The effective degrees of freedom vary with depth but is on average between 40 and 180). The temperature uncertainty increases non-linearly with depth, with the clearly dominant component from a depth larger than 200 m being the parameter estimation through linear regression performed as part of the calibration. The contributions to the combined uncertainty from time-averaging, correction of temperature drift, and bottom temperature equality constraint are negligible. The contribution from the bath temperature uncertainty, the parameter *C*, and the Stokes/anti-Stokes are important at shallow depths (the contribution from the Stokes/anti-Stokes was calculated between averaged periods and does not account for variations within a given averaged period (as these data are not directly available from the DTS instrument)).

These results suggest that calibration should be in focus if one wants to reduce the temperature uncertainty for deeper boreholes (>200 m). Perhaps a way to decrease the calibration contribution to uncertainty is to use more baths, as suggested in [46]. In particular, it would be interesting to investigate the use of a calibration bath at the fiber end to reduce calibration uncertainty. This can be achieved by splicing two fiber channels from the same cable—thereby obtaining a duplexed or double-ended configuration—or by using a single cable to measure the temperature profiles in the annulus and center pipe. Furthermore, the temperature equality prior at the BHE bottom could be kept in such configurations, which could also contribute to reducing the uncertainty (compared to a case without this constraint).

For more shallow boreholes, however, the uncertainty contributions from *C* and the bath temperature should also be considered when trying to reduce the temperature uncertainty. For the bath temperature, a well-mixed bath with a precision thermometer can reduce uncertainty. As for *C*, this could be achieved in similar way through another calibration bath that is better than the internal oven, i.e., where the bath temperature is more uniform and the bath temperature uncertainty is lower.

It is relevant to note that the proposed quantification of uncertainty partly includes potential temperature model errors (i.e., deviations from Equations (Equation 2) and (Equation 3)). This is because model errors will show in the residuals which distribution is partly captured in the linear regression uncertainty. A case for which model error is not fully included is, for instance, when the variance of the residuals is dependent on the distance (in general, any dependency of the residuals on X will invalidate Equation (Equation 14)).

The calibration method and quantification of uncertainty were applied to a specific site with unusual characteristics compared to other BHE applications. In particular, the large borehole depth of 800 m shed light on the different uncertainty components. Applying the methods to another site would nevertheless be needed to test the method’s robustness. A laboratory test under controlled conditions with independent temperature measurements would be even more relevant in that regard. The influence of the choice of distributions in the Monte Carlo study for the bath and bottom conditions should be investigated in future studies. The impact of spatially and temporally varying bath temperature might also be a relevant aspect to consider in such future studies.

One of the main points of quantifying temperature uncertainty with Raman-based DTS is the verification or validation of the borehole heat transfer model. Although it is hard to invalidate a model with field data due to many uncontrolled parameters (e.g., ground water flow, exact borehole geometry) [70], DTS measurements could be very informative as to the model’s strengths and drawbacks. Better models lead to more accurate design and estimation of running costs. In turn, this would help improve the market penetration of environmentally-friendly technologies for heating and cooling (GSHP, BTES, ATES). Another relevant aspect is the determination of the geothermal gradient and related geothermal heat flux as highlighted in [54].

Besides the results strictly related to the DTS calibration and uncertainty, there are some interesting thermal features that deserve further analysis but that can nevertheless be mentioned here. The first one is the heat flux inversion that happens along the BHE depth that can be seen in Figure 5c. At around 200 m depth, the calibrated temperature profiles in the annulus and the center pipe indeed cross each other, meaning no heat is injected (netto) below that point. This intersection moves down the BHE as time elapses under heat injection and reaches around 360 m at the end of the heat injection period. The heat flux inversion has been noticed in previous modeling work [71,72,73] for coaxial BHEs with annulus as the inlet (which is sub-optimal for heat injection). A related question that arises is how much the noticed heat flux inversion would influence the result from a thermal response test (TRT). After all, if no heat injection occurs over a whole borehole section, as is the case here, one may question whether the test can provide any information regarding the thermal parameters within that section.

A second feature that can be noticed is the progressive homogenization of temperature under circulation. Using the mid-depth temperature as the initial temperature uniformly applied in the ground is a longstanding assumption in long-term borehole heat transfer modeling [74]. The more interesting aspect here is perhaps the time it takes for the temperature profile to homogenize. Here, after several hours of circulation, the temperature profile is not yet stabilized. In turn, one may wonder if a long stabilization time would have an impact on heat transfer modeling by, for instance, leading to a different distribution of heat flux along the borehole.

Another notable feature is the temperature disturbances that seem to occur in the topmost 200 m section of the annulus (ch1). This could be due to the laminar flow regime—since that the diameter is larger in that section (165 mm) than in the rest of the borehole (140 mm)—or groundwater inflow, although neither of these explanations is fully satisfactory [66].

## 5. Conclusions

DTS offer interesting perspectives on BHE and GSHP applications; among other things for field tests and heat transfer model validation/verification.

A new calibration method and the associated quantification of uncertainty have been proposed in this paper for single-ended Raman-based DTS. The calibration method is based on a single calibration bath and uses temperature equality at the bottom of the BHE as an extra constraint. In addition, a method to remove fictitious temperature drifts due to ambient air variations was suggested. The different methods were implemented for a case study with a 800 m deep coaxial BHE, in which a DTRT waas conducted.

The calibration method and temperature drift correction show robust features and produce adequate results. The quantification of uncertainty leads to 95% confidence bounds of ±0.58, 0.74, 1.1, and 1.7 K at 200, 300, 500, and 800 m depths, respectively. The temperature uncertainty increases non-linearly with depth and is dominated by the uncertainty in the estimated parameters during the calibration process, for depths larger than 200 m. As for more shallow boreholes, the uncertainty contributions from the parameter *C*, the bath temperature, and the Stokes/anti-Stokes also become important. All of these elements—the calibration process especially—should be focused on if the temperature uncertainty is to be reduced. Such distributed temperature measurements with quantified uncertainty are useful for determining the strengths and drawbacks of heat transfer models and for evaluating local thermal properties. To strengthen the results presented in this study, a test in a laboratory environment with controlled conditions should be performed.

Besides the results strictly related to the calibration and uncertainty quantification, this paper has also offered insights into the thermal features that appear during the DTRT, namely a heat flux inversion along the borehole depth and the relatively slow temperature homogenization under circulation.

## Figures and Tables

**Figure 1 sensors-23-05498-f001:**
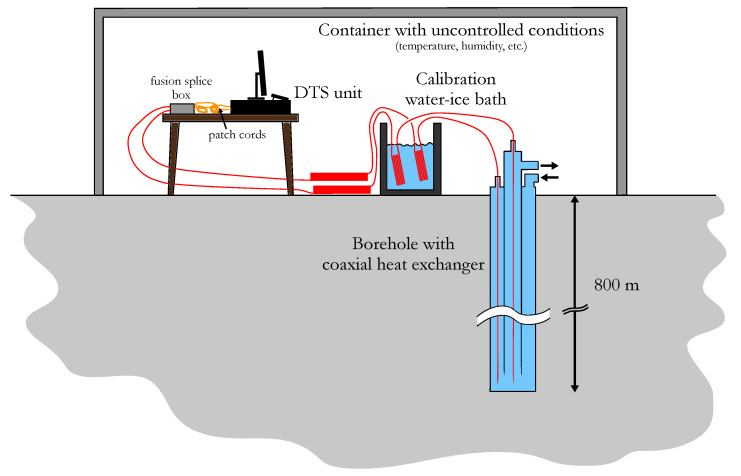
Schematics of the DTS field configuration.

**Figure 2 sensors-23-05498-f002:**
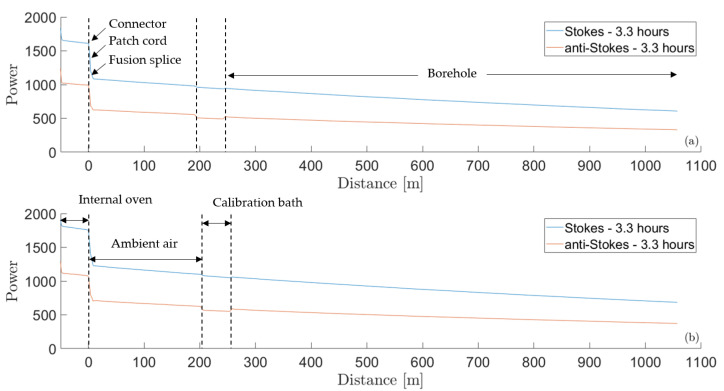
Stokes and anti-Stokes lines and the corresponding sections for the first (**a**) and second (**b**) fiber cables at about 3.3 h after data acquisition start.

**Figure 3 sensors-23-05498-f003:**
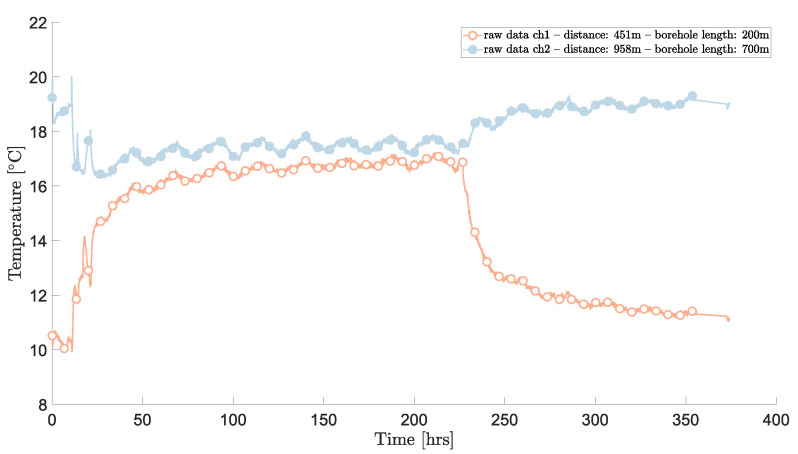
Raw temperature data from two different distances inside the borehole.

**Figure 4 sensors-23-05498-f004:**
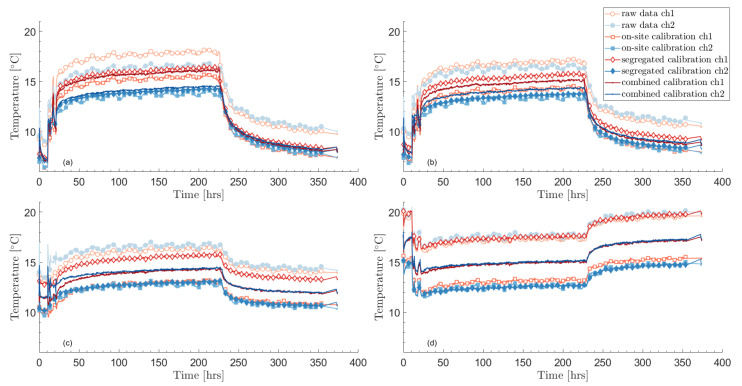
Temperature time profiles after calibration and correction of temperature drifts at four different borehole depths: (**a**) 50 m, (**b**) 150 m, (**c**) 400 m, and (**d**) 750 m.

**Figure 5 sensors-23-05498-f005:**
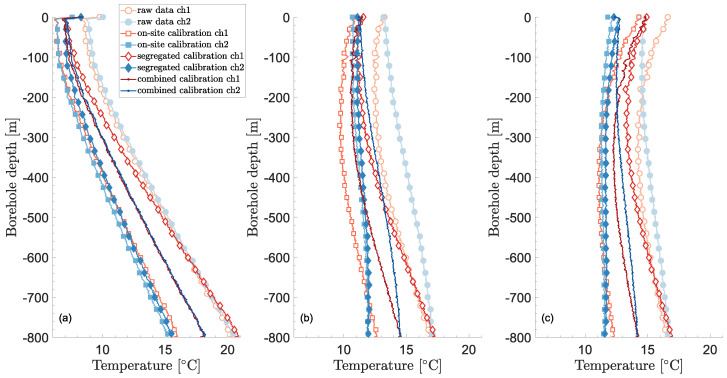
Temperature vertical profiles after calibration and correction of temperature drifts at three different times and conditions: (**a**) 8 hrs—undisturbed, (**b**) 15 hrs—circulation, (**c**) 25 hrs—heat injection.

**Figure 6 sensors-23-05498-f006:**
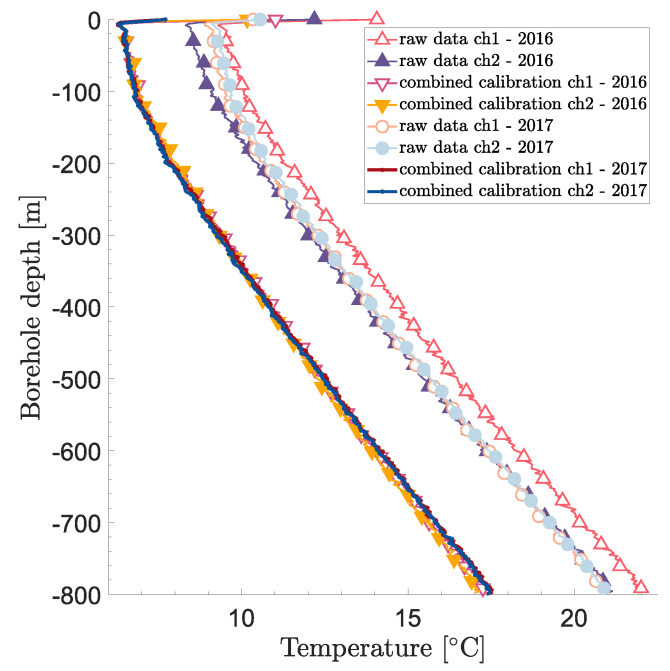
Undisturbed temperature profiles comparison for two different measurements one year apart (2016–2017).

**Figure 7 sensors-23-05498-f007:**
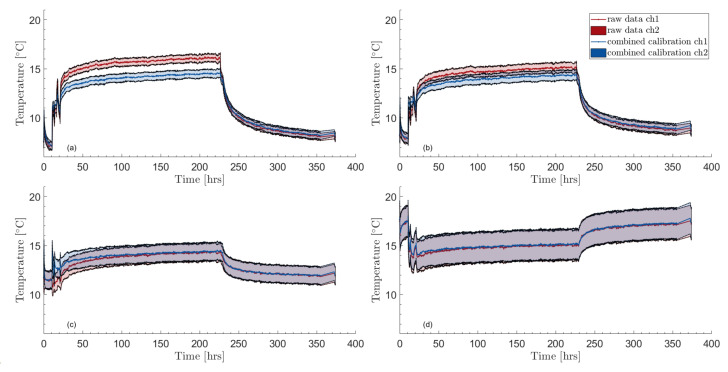
Calibrated temperature time profiles and their 95% uncertainty bounds at four different depths: (**a**) 50 m, (**b**) 150 m, (**c**) 400 m, and (**d**) 750 m.

**Figure 8 sensors-23-05498-f008:**
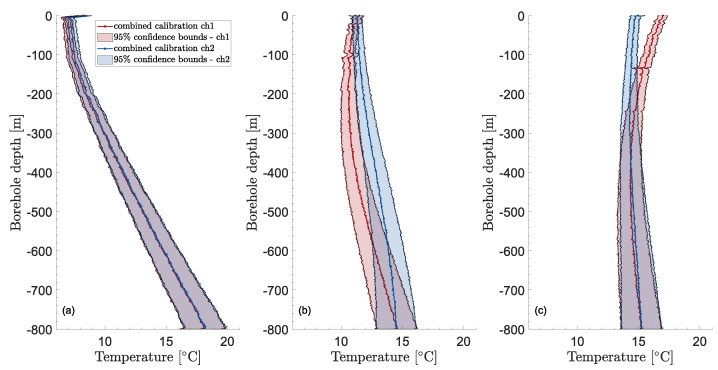
Calibrated temperature vertical profiles and their 95% uncertainty bounds at three different times and conditions: (**a**) 8 h—undisturbed, (**b**) 15 h—circulation, (**c**) 25 h—heat injection.

**Figure 9 sensors-23-05498-f009:**
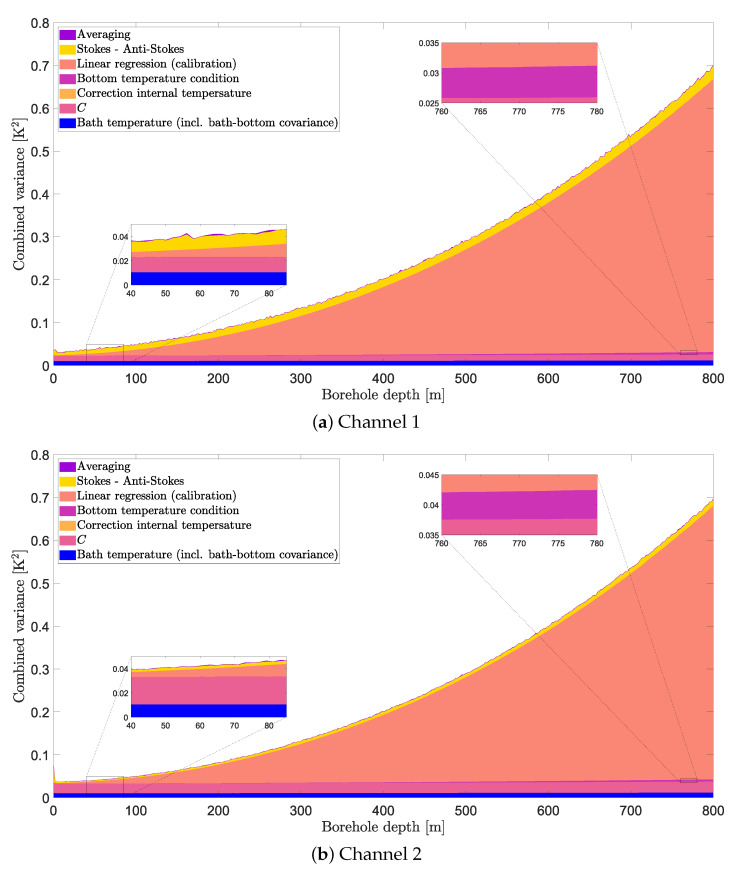
Contribution of each uncertainty component to the combined variance along the borehole depth.

**Table 1 sensors-23-05498-t001:** Parameters obtained after calibration.

	$\Delta \alpha_{1}$	$\Delta R_{1}$	$\Delta \alpha_{2}$	$\Delta R_{2}$
	[m${}^{-1}$]	[-]	[m${}^{-1}$]	[-]
Raw data (default)	$9.8735\times 10^{-5}$	-	$9.8735\times 10^{-5}$	-
On-site calibration	$8.3668\times 10^{-5}$	−0.0106	$7.8139\times 10^{-5}$	−0.0106
Segregated calibration	$11.468\times 10^{-5}$	−0.0165	$7.4989\times 10^{-5}$	−0.0092
Combined calibration	$9.4739\times 10^{-5}$	−0.0121	$9.4630\times 10^{-5}$	−0.0137
Independent combined calibration (one-year interval)	$8.4853\times 10^{-5}$	−0.0104	$8.6884\times 10^{-5}$	−0.0172

**Table 2 sensors-23-05498-t002:** Fixed parameters used during calibration.

γ	$\Delta R_{0}$	$\Delta \alpha_{\mathrm{int}}$	$d_{\mathrm{int}}$	$\beta_{1}$	$\beta_{2}$
[K${}^{-1}$]	[-]	[m${}^{-1}$]	[m]	[K${}^{-1}$]	[K${}^{-1}$]
516.4	−0.0015	$3.9144\times 10^{-5}$	50	$-1.7565\times 10^{-4}$	$-2.9998\times 10^{-4}$

**Table 3 sensors-23-05498-t003:** Time-averaged 95% uncertainty bounds at different borehole depths—sampling interval and temporal averaging of 2.029 m and 5 min.

Depth/Length [m]	0	10	20	100	200	300	400	500	600	700	800
Channel 1 [K]	0.379	0.358	0.383	0.445	0.588	0.742	0.907	1.08	1.27	1.46	1.66
Channel 2 [K]	0.553	0.395	0.400	0.450	0.581	0.735	0.903	1.08	1.27	1.46	1.65
Manufacturer’s “resolution” [69] [K]	0.025	0.025	0.025	0.025	0.027	0.028	0.029	0.030	0.030	0.032	0.033

## Data Availability

The data presented in this study are available on request from the corresponding author.

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
