# Peer review of "Calibration and Uncertainty Quantification for Single-Ended Raman-Based Distributed Temperature Sensing: Case Study in a 800 m Deep Coaxial Borehole Heat Exchanger"

_sensors, 2023, doi:10.3390/s23125498_

Round 1
Reviewer 1 Report
Commercial Raman-DTS systems are not adequate in addressing various calibration challenges and uncertainties in measurements inherent to several factors such as instrument types, calibrations baths, fusion splices and dynamic ambient conditions, usually the scenario in field-tests environment. As authors pointed out, the researchers are developing and implementing different ways of calibration to increase the accuracy and precision of the spatial temperature measurement with different degree of success on various case studies. Here the authors present the new calibration method to estimate the measurement uncertainty more accurately for a single ended DTS configurations using only one calibration bath implemented in temperature measurement in a 800 m deep coaxial BHE. Though the calibration method is robust for the very case study in the same BHE data, its versatility has not been tested against any other test environments. It looks unlikely to have a universal calibration method applicable for different applications because of the case specific uncertainty parameters. Nevertheless, the authors meticulously described the calibration model and improved the accuracy of the temperature measurement for BHE applications which other application areas can benefit from the insights.
With some typos corrections in the manuscripts, the article has cemented its case to be published in sensors journal.
Author Response
Please see the attachement.

Reviewer 2 Report
I have reviewed the manuscript entitled "Calibration and Uncertainty Quantification for Single-Ended Raman-Based Distributed Temperature Sensing: Case Study in a 800 m Deep Coaxial Borehole Heat Exchanger" carefully. The authors proposed a new calibration method for single-ended Raman-based Distributed Temperature Sensing (DTS) configurations in addition to a method to remove fictitious temperature drifts caused by ambient air. The authors did an experimental Distributed Thermal Response Test (DTRT) case study in an 800 m deep coaxial Borehole Heat Exchanger (BHE) and their results showed that their methods have features and give adequate results. The results are strictly related to the DTS calibration and uncertainty. Furthermore, the authors’ experimental results give many insights into thermal features. The authors provide a lot of useful experimental data, and an in-depth analysis and many interesting conclusions. I think the paper can be accepted after some typos such as
Page 4:
2.2
The fibers are 166 inserted into a 800 m deep water-filled borehole equipped with a coaxial BHE (see [? ] for 167 more details on the BHE).
Reviewer 3 Report
The paper sound interesting. The authors should revise the paper according to the following major comments:
- The abstract is too long and should be shortened. The main aim of the abstract is to present the highlight of the research. The main results of the research and the different methods to achieve these results.
- The introduction section is very comprehensive and well presented
- Nomenclature must be added to the paper as well as mathematical/physical units of each parameter and variable.
- In eq. 4 the beta is a function of T or it is multiplication by beta?
- The analysis of the results present in the figures should be extended.
- In eq 9, 10, 17... the above line of the mathematical expression, is the conjugate operator?
- The following papers can be added to the current research:
1: Nave, O., Hareli, S., & Gol’dshtein, V. (2014). Singularly perturbed homotopy analysis method. In Applied Mathematical Modelling (Vol. 38, Issues 19–20, pp. 4614–4624). Elsevier BV. https://doi.org/10.1016/j.apm.2014.03.013
2: Luo, Y., Guo, H., Meggers, F., & Zhang, L. (2019). Deep coaxial borehole heat exchanger: Analytical modeling and thermal analysis. In Energy (Vol. 185, pp. 1298–1313). Elsevier BV. https://doi.org/10.1016/j.energy.2019.05.228
- Please explain how did you obtain the mathematical expression present in eq 19.
- The section result, discussion, and conclusion present very well.
Reviewer 4 Report
The calibration method proposed is interesting, and the paper is well-structured and well-presented.
However, the experiment should be conducted in two environments:
Lab environment: using a container in which the conditions are controlled, discarding the influence of daily temperature drifts, in order to test the method's robustness and the results of the top 10-15 m of the borehole length doesn’t need to be excluded (lines 417-418, lines 439-440).
Real environment: using a container in which the conditions aren’t controlled, in order to apply the correction for fictitious temperature drifts caused indirectly by ambient temperature variations.
In addition, please update the references. In fact, about 1/3 of references are more than 10 years old.
Minor improvements:
Line 6: Please replace “…Distributed Thermal Response Test…” by “…Distributed Thermal Response Test (DTRT)…”
Line 15: Please review the Keywords avoiding acronyms and generic words such as “Fiber”, “Coaxial” and “Deep”
Line 66: Please explain the meaning of RMSE
Line 95: Please replace “Des Tombes et al. [45] propose…” by “Des Tombe et al. [45] propose…”
Line 167: Please review “see [? ]”
Line 173: Please replace “Nylon” by “nylon”
Line 235: Please replace “…missing).s” by “…missing).”
Line 379: Please replace “…shown in 3.” by “…shown in Figure 4.”
Table 3: Please locate it after line 446
Figure 9: Please locate it after line 460
Line 507: Please replace “…condition.” by “condition,”
Line 582: Please replace “BHe” by “BHE”
Line 599: Please review “satisfactory [? ]”
Figures 4, 5, 6, 7, and 8: Please use contrasting colors to highlight the differences
Round 2
Reviewer 3 Report
I can confirm that the authors revise the paper according to my comments.
Author Response
The authors would like to thank the reviewer once more for their comments and suggestions.
Reviewer 4 Report
I would like to thank the authors for the cover letter and the improvements performed in this second version. Concerning the paper, although improving references, the leading paper fragility remains – the absence of an experiment conducted in a lab environment.
In my opinion, the presented case isn’t sufficiently robust to be published. The authors must strengthen the research through an experiment in a controlled environment, discarding the influence of daily temperature drifts. Or at least the limitations of the study must be mentioned in sections 4 and 5 in this second version. In fact, I can't find in the paper the statements regarding the research limitations: “Applying the methods to another site would nevertheless be positive to test the method robustness. A lab test with independent temperature measurements would be even more relevant.”
